# The Effect of miR-146a on the Gene Expression of Immunoregulatory Cytokines in Human Mesenchymal Stromal Cells

**DOI:** 10.3390/ijms21186809

**Published:** 2020-09-16

**Authors:** Jung Hwa Ko, Joo Youn Oh

**Affiliations:** 1Department of Ophthalmology, Seoul National University Hospital, 101 Daehak-ro, Jongno-gu, Seoul 03080, Korea; kjh382@hanmail.net; 2Department of Ophthalmology, Seoul National University College of Medicine, 103 Daehak-ro, Jongno-gu, Seoul 03080, Korea

**Keywords:** immunoregulatory factor, macrophage, mesenchymal stem/stromal cell, microRNA, miR-146a

## Abstract

Mounting evidence indicates that microRNAs (miRNAs), including miR-146a, have an impact on the immunomodulatory activities of mesenchymal stem/stromal cells (MSCs). Suppression of inflammatory macrophage activation is one of the main immunomodulatory mechanisms of MSCs. Here, we investigated whether miR-146a in MSCs might play a role in the effects of MSCs on macrophage activation. A miRNA microarray revealed that miR-146a was the most highly upregulated miRNA in MSCs upon co-culture with activated macrophages. Inhibition of miR-146a in MSCs through miR-146a inhibitor transfection had a different effect on the expression of immunoregulatory factors secreted by MSCs. Pentraxin 3, tumor necrosis factor-inducible gene 6, and cyclooxygenase-2, which are well-known mediators of the immunomodulatory functions of MSCs, were significantly upregulated in MSCs after miR-146a knockdown. By contrast, hepatocyte growth factor and stanniocalcin 1, other immunoregulatory molecules expressed by MSCs, were downregulated by miR-146a knockdown. Consequently, the inhibition of miR-146a in MSCs did not change the overall effect of MSCs on the suppression of inflammatory macrophage activation or the induction of anti-inflammatory macrophage polarization.

## 1. Introduction 

MicroRNAs (miRNAs) are small non-coding RNAs that regulate the expression and function of their target mRNAs at a post-transcriptional level, thereby playing an important role in diverse cellular processes [1]. Recent studies have shown that miRNAs regulate and mediate the immunomodulatory functions of mesenchymal stem/stromal cells (MSCs) [2,3]. MSCs, multipotent adult stromal cells originally found in the bone marrow (BM) [4], have been actively investigated as a promising candidate for cell therapy for a number of immune-mediated disorders, due to their potent immunomodulatory activities. MSCs exert their immunosuppressive effects largely through the modulation of macrophage activation via paracrine mechanisms [5,6].

In this study, from miRNA microarrays, we identified miR-146a as the most strongly upregulated miRNA in human BM-derived MSCs following co-culture with activated macrophages. To evaluate the effects of miR-146a on the immunomodulatory activities of MSCs, we investigated the transcriptome of MSCs after miR-146a knockdown and analyzed the expression of immunoregulatory factors. In addition, we evaluated the effects of MSCs with miR-146a knockdown on macrophage activation.

## 2. Results

### 2.1. The MiRNA Expression Profile of MSCs Co-Cultured with Activated Macrophages

In a previous study, we found that MSCs inhibited the inflammatory activation of macrophages in response to serial stimulation by lipopolysaccharide (LPS)/ adenosine triphosphate (ATP), and induced polarization of macrophages into anti-inflammatory phenotypes [7]. It has been reported that miRNAs play a critical role in the regulation of the immunomodulatory activities of MSCs [3]. Hence, in the present study, we analyzed the expression profiles of miRNAs in MSCs using miRNA microarrays following an 18-hour co-culture with LPS/ATP-stimulated macrophages (Figure 1A; raw data deposited at ArrayExpress under accession number E-MTAB-9418). As a result, a total of 11 miRNAs were found to be upregulated, showing a >three-fold increase, and five miRNAs were downregulated, showing a >three-fold decrease, in MSCs co-cultured with LPS/ATP-stimulated macrophages compared with MSCs cultured alone (Figure 1B). Among differentially expressed miRNAs, the most highly upregulated was miR-146a (nine-fold increase) (Figure 1B). We validated the microarray data using real-time reverse transcription polymerase chain reaction (RT-PCR), and detected an approximately five-fold increase in miR-146a expression in MSCs following co-culture with LPS/ATP-stimulated macrophages (Figure 1C).

### 2.2. Transcriptome Profile of MSCs with miR-146a Inhibition 

To investigate the effects of miR-146a on the immunomodulatory properties of MSCs, we transfected MSCs with either a miR-146a inhibitor (miR-146a inhibitor-MSCs) or a negative control (NC-MSCs), and cultured the cells together with LPS/ATP-stimulated macrophages in a transwell system for 18 hours (Figure 2A). Then, we performed RNA sequencing on the MSCs and compared the transcriptome of miR-146a inhibitor-MSCs with that of NC-MSCs (Figure 2B). In particular, we analyzed the immune response- and secretion-related genes, because MSCs control immune responses largely through the secretion of paracrine factors [5,6,7]. The data revealed that 10 genes were upregulated >three-fold and six genes downregulated >three-fold in miR-146a inhibitor-MSCs, compared to NC-MSCs (Figure 2B). The upregulated genes included pentraxin 3 (PTX3), tumor necrosis factor-inducible gene 6 (TSG-6), and cyclooxygenase-2 (COX-2), which are well-known to mediate the immunomodulatory actions of MSCs [7,8,9,10,11,12]. Meanwhile, hepatocyte growth factor (HGF) and stanniocalcin 1 (STC-1), other therapeutic factors expressed by MSCs [13,14,15], were among the downregulated genes in miR-146a inhibitor-MSCs. The expression of these immunoregulatory molecules was further validated by real-time RT-PCR and enzyme-linked immunosorbent assay (ELISA) (Figure 2C,D). Together, the results confirmed that miR-146a inhibition in MSCs had opposite effects on the expression of immunoregulatory molecules, by enhancing PTX3, TSG-6, and COX-2, and by repressing HGF and STC-1.

### 2.3. Effects of miR-146a Inhibition in MSCs on the Modulation of Macrophage Polarization 

We next investigated the functional significance of miR-146a in terms of the regulatory effects of MSCs on macrophage activation. Macrophages were stimulated with LPS/ATP and co-cultured with either miR-146a inhibitor-MSCs or NC-MSCs. Eighteen hours later, the macrophages were subjected to analysis (Figure 3A). Both miR-146a inhibitor-MSCs and NC-MSCs significantly decreased the mRNA levels of pro-inflammatory cytokines (TNF-α, IL-12a, and IL-12b) in LPS/ATP-activated macrophages, as measured by real-time RT-PCR (Figure 3B). The surface expression of HLA-DR, CD11b, and co-stimulatory molecule CD80 in LPS/ATP-activated macrophages was repressed similarly by miR-146a inhibitor-MSCs and NC-MSCs, as analyzed by flow cytometry, whereas the expression of anti-inflammatory marker CD206 was increased (Figure 3C). In addition, miR-146a inhibitor-MSCs and NC-MSCs were equally effective at inducing the secretion of the immunoregulatory cytokines amphiregulin (AREG), IL-10, TGF-β1, and TGF-β2, as measured by ELISA (Figure 3D). Therefore, the data clearly demonstrated that miR-146a inhibition in MSCs did not affect the activity of MSCs in terms of the suppression of pro-inflammatory activation of macrophages, or in terms of direct macrophage polarization into anti-inflammatory phenotypes.

## 3. Discussion

MiR-146a is induced in various types of immune cells and adult stem cells, including monocytes and MSCs, in response to inflammatory priming in an NF-κB-dependent manner [2,3,16,17,18,19]. In monocytes and macrophages, miR-146a has been shown to negatively regulate inflammatory activation by blocking NF-κB activation [20], decreasing TRAF6 and IRAK1 expression [17], driving polarization into anti-inflammatory M2 phenotypes [21], and contributing to endotoxin-induced tolerance [22,23]. On the other hand, there are contradictory data on the role of miR-146a in the immunomodulatory activities of MSCs. An elegant study by Matysiak et al. [19] showed that miR-146a was upregulated in BM-derived MSCs during neuronal differentiation, and increased miR-146a expression in MSCs abrogated the immunoregulatory effects of MSCs by inhibiting the synthesis of PGE2, an important therapeutic factor secreted by MSCs. In contrast, other studies have suggested that miR-146a is embedded in MSC exosomes and transferred to immune cells, such as macrophages and group 2 innate lymphoid cells, where it suppresses inflammatory activation of immune cells, thereby mediating the immunomodulatory functions of MSCs [24,25].

In line with previous studies, we observed that miR-146a was strongly upregulated in MSCs following stimulation by activated macrophages. However, in our setting, MSCs with miR-146a inhibition were as effective in suppressing macrophage activation and inducing anti-inflammatory macrophage phenotypes as control MSCs, an indication that miR-146a did not affect the action of MSCs in modulating macrophage polarization. This finding can be explained, at least in part, by our observation that miR-146a had differential effects on the expression of various immunoregulatory molecules in MSCs. Consistent with a study by Matysiak et al. [19], our transcriptome analysis revealed that miR-146a inhibition enhanced COX-2 in MSCs, an essential enzyme in PGE2 synthesis, as well as PTX3 and TSG-6. Simultaneously, miR-146a knockdown repressed other immunoregulatory factors, specifically HGF and STC-1, in MSCs. Since the immunomodulatory activities of MSCs are mediated through the combined actions of these paracrine factors, it is likely that miR-146a inhibition does not change the overall effects of MSCs on macrophage polarization. 

Of note, both miR-146a and the immunoregulatory molecules that were negatively regulated by miR-146a (PTX3, TSG-6, and COX-2) were all increased in MSCs following macrophage co-culture. This suggests that miRNAs other than miR-146a play a more important role in the regulation of PTX3, TSG-6, and COX-2 expression in MSCs. Our group recently showed that the immunoregulatory function of MSC-derived extracellular vesicles in LPS-stimulated splenocytes was positively correlated with their PTX3, let-7b, and miR-21 levels [9]. Another recent study by Miyaji et al. revealed that TSG-6 expression in BM-derived MSCs was upregulated by miR-23b-39, miR-204-3p, miR-1247-3p, and miR-326-5p in the cells [26]. Further studies would be needed to identify miRNAs that directly regulate the expression of PTX3, TSG-6, and COX2 in MSCs. 

Apart from its role in the immunomodulatory action of MSCs, miR-146a has been shown to be involved in the proliferation, differentiation, and senescence of MSCs [18,27,28,29,30]. Thus, it would be interesting to investigate the effects of macrophage-induced miR-146a expression in MSCs on the survival and stemness of cells.

## 4. Materials and Methods

### 4.1. Cell Culture and Transfection 

Human BM-derived MSCs (donor #8004L) were provided by the Center for the Preparation and Distribution of Adult Stem Cells. Passage 2 MSCs were cultured at 5% CO_2_ and 37 °C in complete culture media (CCM), consisting of α-minimal essential medium (Gibco, Thermo Fisher Scientific, Waltham, MA, USA), 17% fetal bovine serum (FBS, Gibco, Thermo Fisher Scientific, Waltham, MA, USA), 1% penicillin-streptomycin (PS, Lonza, Walkersville, MD, USA), and 2 mM L-glutamine (Lonza, Walkersville, MD, USA). Prior to macrophage co-culture, MSCs were seeded in transwell inserts (Millicell^®^ Cell Culture Inserts, Merck Millipore, Darmstadt, Germany) at a ratio of 1:5 (MSCs: macrophages), and were incubated in CCM for 24 hours. Then, the MSCs in transwell inserts were co-cultured with macrophages in macrophage culture media for 18 hours.

For miR-146a knockdown, MSCs were transfected with either miR-146a inhibitor (has-miR-146a-5p, Thermo Fisher Scientific, Waltham, MA) or a negative control (Invitrogen, Thermo Fisher Scientific, Waltham, MA, USA) using Lipofectamine™ RNAiMAX Transfection Reagent (Invitrogen, Thermo Fisher Scientific, Waltham, MA, USA), according to the manufacturer’s instructions.

Macrophages were differentiated from human peripheral blood-derived monocytes (Seoul National University Hospital cell bank, Seoul, Korea) by treatment with 300 ng/ml phorbol 12-myristate-13-acetate (InvivoGen, San Diego, CA, USA) for 3 hours, and were sequentially stimulated by 2 μg/mL LPS (InvivoGen) for 3 h and 5 mM ATP (InvivoGen) for 45 minutes. After washing with phosphate buffered saline (PBS, Welgene, Daegu, Korea), the macrophages were cultured in RPMI1640 media (Welgene, Daegu, Korea) containing 2% (*v*/*v*) heat-inactivated FBS (Gibco, Thermo Fisher Scientific, Waltham, MA, USA) and 1% PS (Lonza, Walkersville, MD, USA) at 5% CO_2_ and 37 °C.

### 4.2. RNA Sequencing 

The RNeasy Mini Kit (Qiagen, Hilden, Germany) or the miRVana miRNA Isolation Kit (Invitrogen) were used to isolate total RNA; 500 ng total RNA was prepared from MSCs, and the construction of the library was made with the QuantSeq 3’ mRNA-Seq Library Prep Kit (Lexogen, Inc., Vienna, Austria) according to the manufacturer’s instructions. High-throughput sequencing was performed as single-end 75 sequencing using NextSeq 500 (Illumina, Inc., San Diego, CA, USA). Gene classification was based on searches of the DAVID and Medline databases.

### 4.3. Real-Time RT-PCR

First-strand cDNA was synthesized by reverse transcription using a High Capacity RNA-to-cDNA™ Kit (Applied Biosystems, Carlsbad, CA, USA), and real-time amplification for a specific gene was carried out with TaqMan^®^ Universal PCR Master Mix (Applied Biosystems, Carlsbad, CA, USA) in an automated instrument (ABI 7500 Real Time PCR System, Applied Biosystems, Carlsbad, CA, USA). Data were normalized to GAPDH and expressed as fold changes relative to controls. All PCR probe sets were purchased from Applied Biosystems (TaqMan^®^ Gene Expression Assay kits, Carlsbad, CA, USA).

### 4.4. ELISA

The concentrations of TSG-6, STC-1, HGF, AREG, IL-10, active TGF-β1, and active TGF-β2 were assayed in the cell-free culture supernatants using DuoSet^®^ ELISA kits (R&D Systems, Minneapolis, MN, USA).

### 4.5. Flow Cytometry 

The cells were stained with fluorescence-conjugated antibodies against HLA-DR, CD80, CD11b, and CD206 (eBioscience, San Diego, CA, USA), and fluorescence was measured using an S1000EXi Flow Cytometer (Stratedigm, San Jose, CA, USA). Data were analyzed using the FlowJo program (Tree Star, Inc., Ashland, OR, USA).

### 4.6. Statistical Analysis

All data were analyzed using one-way ANOVA followed by Tukey’s multiple comparisons tests. Statistical analysis and graphical generation of data were done with GraphPad Prism^®^ software 8.4.2 (San Diego, CA, USA). The data are presented as the mean ± SD, and differences were considered significant at *p* < 0.05.

## 5. Conclusions

In summary, our study demonstrated that miR-146a in MSCs was strongly induced in response to co-culture with activated macrophages. Inhibition of miR-146a in MSCs had different effects on the secretion of immunoregulatory factors by upregulating PTX3, TSG-6, and COX-2 and by downregulating HGF and STC-1. Despite the change in miR-146a expression, the capacity of MSCs to control excessive activation of macrophages was robustly maintained. 

## Figures and Tables

**Figure 1 ijms-21-06809-f001:**
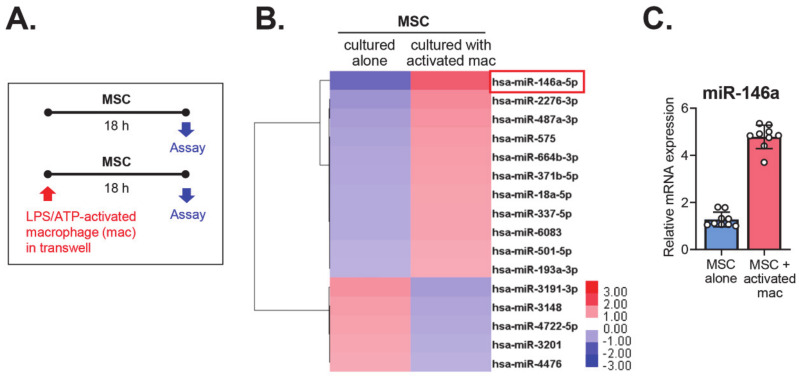
The miRNA changes in mesenchymal stem/stromal cells (MSCs) after co-culture with activated macrophages. (**A**). Experimental scheme. MSCs were cultured alone or co-cultured with macrophages activated by lipopolysaccharide (LPS) and adenosine triphosphate (ATP). Eighteen hours later, the MSCs were evaluated by miRNA microarrays and real-time reverse transcription polymerase chain reaction (RT-PCR). (**B**). Heat-map of microarray assays for miRNAs that were up- or downregulated >three-fold in MSCs co-cultured with activated macrophages, as compared to MSCs cultured alone. (**C**). The level of miR-146a in MSCs cultured with activated macrophages relative to MSCs cultured alone, as determined by real-time RT-PCR.

**Figure 2 ijms-21-06809-f002:**
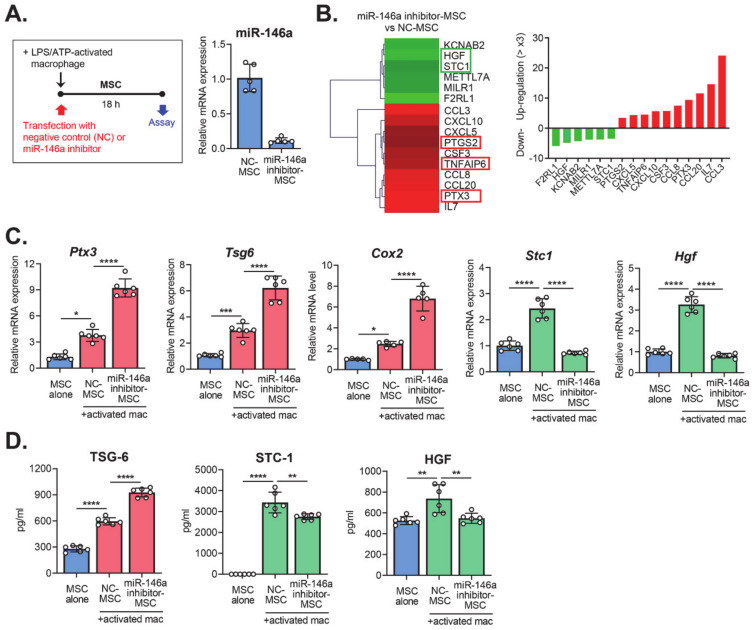
The changes in the expression of immunoregulatory molecules in MSCs after miR-146a knockdown. (**A**). Scheme for experiments. MSCs were transfected with a miR-146a inhibitor (miR-146a inhibitor-MSC) or a negative control (NC-MSC) and were co-cultured with LPS/ATP-activated macrophages for 18 hours. The knockdown of miR-146a in MSCs was confirmed by real-time RT-PCR. (**B**). A heat-map and list of immune response- and secretion-related genes based on RNA sequencing of miR-146a inhibitor-MSCs as compared with NC-MSCs. Shown are the genes whose expressions were up- or downregulated by >three-fold in miR-146a inhibitor-MSC relative to NC-MSC. (**C**). Real-time RT-PCR for immunoregulatory genes in MSCs. The mRNA levels of each gene in MSCs co-cultured with LPS/ATP-activated macrophages are shown as the relative values to those in MSCs cultured alone. (**D**). ELISA (enzyme-linked immunosorbent assay) for secreted protein levels of immunoregulatory molecules. * *p* < 0.05, ** *p* < 0.01, *** *p* < 0.001, and *****p* < 0.0001, as analyzed by one-way ANOVA and Tukey’s multiple comparisons test.

**Figure 3 ijms-21-06809-f003:**
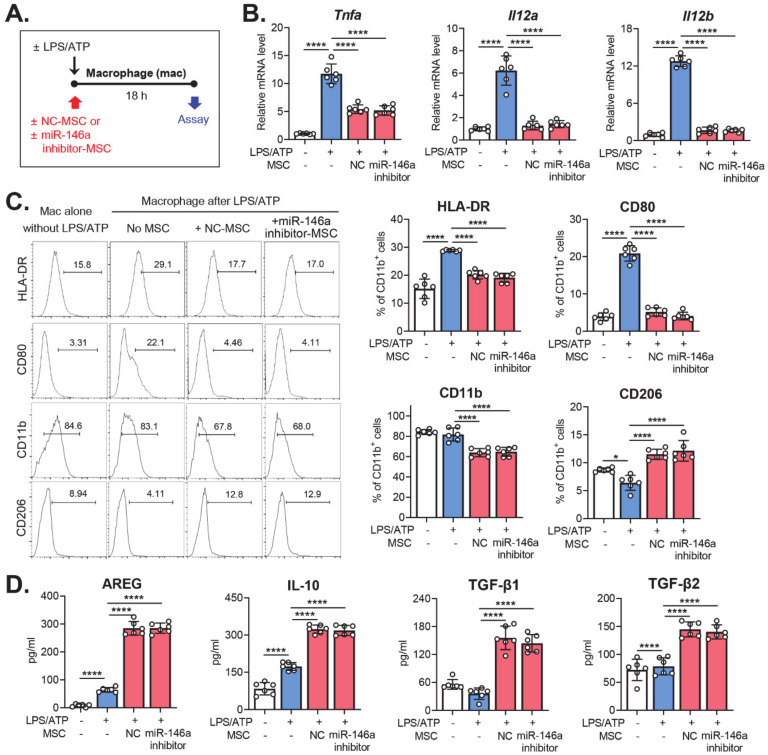
Absence of effects of miR-146a inhibition on the activity of MSCs in terms of modulating macrophage polarization. (**A**). Scheme for experiments. Macrophages were activated by serial treatment with LPS and ATP and then co-cultured with MSCs that had been transfected with a miR-146a inhibitor (miR-146a inhibitor-MSC) or a negative control (NC-MSC). Eighteen hours later, the macrophages were evaluated for inflammatory activation status. (**B**). Real-time RT-PCR for pro-inflammatory cytokines in macrophages. Shown are the mRNA levels in LPS/ATP-activated macrophages relative to LPS/ATP-untreated macrophages. (**C**). Flow cytometric analysis for surface expression of the MHC class II molecules HLA-DR, CD80, CD11b, and CD206 on macrophages. (**D**). ELISA of the secreted levels of immunoregulatory proteins in the cell-free culture supernatant. **** *p* < 0.0001, as analyzed by one-way ANOVA and Tukey’s multiple comparisons test.

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
