# Peer review of "The Effect of miR-146a on the Gene Expression of Immunoregulatory Cytokines in Human Mesenchymal Stromal Cells"

_ijms, 2020, doi:10.3390/ijms21186809_

Round 1

Reviewer 1 Report

I recommend that it should be revised taking into account the changes requested.

The reviewed manuscript ijms-896237 "Opposite effects of miR-146a on gene expression of immunoregulatory cytokines in human mesenchymal stromal cells" describes a study of immune regulation by BM-MSCs under inflammatory conditions. All experimental in the manuscript are described and well paned with all necessary control, the conclusion totally correspond the data. Although the topic is of interest and this manuscript was written well, some minor concern raised by the reviewer need to be addressed.

    1. How to choose the culture time for co-culture? Whiter effects have time-dependent manner? Please discuss.
    2. How many use MSCs lot number in this research.
    3. For macrophage polarization, could the author evaluated macrophage maker (M1: CD11b, M2: CD163 or CD206) to further verify their finding?

Author Response

  1. How to choose the culture time for co-culture? Whiter effects have time-dependent manner? Please discuss.
  • We understand the reviewer’s concern about the culture time because LPS can induce either inflammatory activation or tolerance in macrophages in dose- and time-dependent manners. In this experiment, we chose the culture time of ’18 hours’ based on our previous studies (Reference 7 and 15) which revealed that LPS-ATP serial stimulation activate inflammatory polarization of macrophages as well as inflammasome activation when assessed from 2 to 24 hours of stimulation (we did not evaluate time points later than 24 hours). In our experience, cellular ROS was most highly upregulated at 2 to 4 hours as shown in Reference 15, and inflammatory molecule expression was highly increased at 18 to 24 hours as shown in Reference 7 (we referred to the two references in the manuscript). Since we did not perform cellular ROS measurement in this study, we included only 18 hour-time point data.
  1. How many use MSCs lot number in this research.
  • We used MSCs derived from one donor (#8004L) and mentioned this in the revised manuscript (Method section, line 159).
  1. For macrophage polarization, could the author evaluated macrophage maker (M1: CD11b, M2: CD163 or CD206) to further verify their finding?
  • As requested, we included additional macrophage marker data (CD11b and CD206) in the revised Figure 3 (line 103-106, 120, 199-200).

Reviewer 2 Report

The paper is very good, of high scientific interest and opens new scenarious in understanding the physiologic role of MSCs. It is important that the authors have checked the role of microRNA146a in the infiammatory activation of macrophages. Up to now, there are no studies that investigated the connection between this miRNA of MSCs and macrophages.

The mansucript is very interesting and well organized. The results are well written and clear in each part. The figures are good. In my opinion, the authors must confirm molecular data with corresponding expression of proteins, at least for the main genes.

Author Response

In response, we performed ELISA to measure the protein levels of TSG-6, HGF and STC-1 in the cell-free culture supernatants and included the results in Figure 2D (Line 81, 94-95, 194). 

Reviewer 3 Report

In the current study, Ko and Oh studied the effects of miR-146a on gene expression of immunoregulatory cytokines in human mesenchymal stromal cells co-cultured with activated macrophages. The authors proposed that miR-146a has opposite effects; however, it doesn’t make sense for me for the following reason. In Figure 2C, all of the genes shown here were upregulated in NC-MSC co-cultured with activated macrophages compared with MSC alone, which is thought to be induced by the upregulation of by miR-146a. If these genes are direct targets of miR-146a, miR-146a inhibitor should work similarly to these genes. However, some genes were upregulated but the others were downregulated by the inhibitor, suggesting that the upregulation of Ptx3, Tsg6, and Cox2 was not caused by any other reasons, including other miRs, than miR-146a upregulation. If this is the case, miR-146a itself does not have opposite effects.

Author Response

We understand that it is counterintuitive that both miR-146a and its negative targets Ptx3, Tsg6 and Cox2 are increased in MSCs by macrophage coculture. As the reviewer wisely speculated, this finding indicates that  Ptx3, Tsg6 and Cox2 might not be direct targets of miR-146a and regulated by other miRNAs. We added this spculation in the discussion (line 148-151). 

Round 2

Reviewer 3 Report

Unfortunately, the authors did not properly respond to my previous comments. The current results do not support the conclusion.

Author Response

We understand the Reviewer's concern in that we could not identify the miRNA(s) positively regulating the expression of COX-2, TSG-6 and PTX3; the inhibition of miR-146a upregulated COX-2, TSG-6 and PTX3, whereas MSCs with increased miR-146a still had the upregulation of COX-2, TSG-6 and PTX3. Identification of other miRNAs than miR-146a that regulate individual genes in MSCs is beyond the subject of the present study, and in this study, we focused on the effect of miR-146a which was identified as the most highly upregulated in microarrays of MSCs cocultured with activated macrophages.

Upon fully acknowledging the concern, we removed the term "opposite" in the title and changed to "different" in the text, and we discussed the candidate miRNAs that might be involved in the regulation of PTX3 and TSG-6 in MSCs based on the previous studies and added the references in the Discussion section (Line 154-158). We could not find the study showing the miRNAs in MSCs positively regulating COX-2 and PGE2 expression; therefore, we added the sentence "Further studies would be needed to identify miRNAs that directly regulate the expression of COX2 in MSCs" (Line 158-159).